# Dietitians’ Experiences of Providing Oral Health Promotion to Clients with an Eating Disorder: A Qualitative Study

**DOI:** 10.3390/ijerph192114193

**Published:** 2022-10-30

**Authors:** Tiffany Patterson-Norrie, Lucie Ramjan, Mariana S. Sousa, Ajesh George

**Affiliations:** 1Australian Centre for Integration of Oral Health, School of Nursing and Midwifery, Western Sydney University, Penrith, NSW 1871, Australia; 2Ingham Institute for Applied Medical Research, Liverpool, NSW 2170, Australia; 3Translational Health Research Institute, Western Sydney University, Campbelltown, NSW 2560, Australia; 4IMPACCT—Improving Palliative, Aged and Chronic Care through Clinical Research and Translation, Faculty of Health, University of Technology Sydney, Broadway, NSW 2007, Australia; 5School of Dentistry, Faculty of Medicine and Health, University of Sydney, Surry Hills, NSW 2010, Australia

**Keywords:** feeding and eating disorders, dietetics, dietitian, oral health promotion, early intervention, non-dental health professionals

## Abstract

(1) Background: Eating disorders (EDs) can seriously impact oral health, leading to irreversible dental damage. Dietitians play a key role in the care of people with an ED and are well-placed to promote oral health. However, there is currently little understanding of how dietitians perceive their role in this space. This study aimed to explore the perceptions and role of dietitians in providing oral health promotion to their clients in an ED clinical setting. (2) Methods: This descriptive qualitative study used semi-structured interviews to explore the perceptions of 14 registered dietitians practicing across seven states in Australia. Participants were recruited using a combination of purposive and snowball sampling. A hybrid thematic analysis approach was undertaken to identify and describe the key themes generated from the data. (3) Results: Generally, dietitians were insightful and knowledgeable of the oral health issues that clients with an ED may be experiencing. However, dietitians’ practices across education, screening, and referrals were inconsistent. Challenges such as inadequate training, unknown referral pathways, and clear guidelines were cited as significant barriers to practice. (4) Conclusions: The results reinforce the need for education and the development of targeted strategies that address challenges to oral health promotion in dietetic practice.

## 1. Introduction

Eating disorders (EDs) are a psychopathological condition characterised by an unhealthy preoccupation with food and body image leading to inappropriate eating patterns and/or self-harm [1]. Compromised nutritional intake and starvation, through restrictive dieting, bingeing, and purging can lead to a myriad of multi-systemic complications, including impaired cardiovascular function, muscle wasting, bone demineralisation, and electrolyte imbalances as well as irreversible damage to oral health [2,3,4,5].

The impact of eating disorders (EDs) on oral health is less well-known. Often, oral health changes begin as subtle manifestations in the mouth; however, years of bingeing and purging can lead to irreversible dental erosion or decay [6,7,8]. Two systematic reviews and a meta-analysis exploring the relationship between eating disorders and oral health indicated an increased risk of tooth erosion (OR = 12.4, 95%CI = 4.1–37.5) and higher rates of decayed, missing, or filled teeth surfaces than the controls [9,10] regardless of ED subtype [8]. Further, the impact of compensatory behaviours, such as mechanical trauma, excessive intake of deleterious nutrients, or metabolic impairment related to a nutritional deficiency, precipitates changes in the gum, including periodontal disease and changes to salivary amylase, leading to inflammation or breakdown of the oral mucosa [11,12]. Given good oral health is essential for adequate mastication, speech, and appearance, compromised oral health can perpetuate ED-associated behaviours such as food avoidance and impact treatment progress [13].

Considering this, people with an ED may be reluctant to seek assistance from dental services. Individuals with an ED undergoing dental treatment were observed to exhibit greater levels of anxiety and fear than those without an ED because of shame, embarrassment, and hypersensitivity to auditory and contact stimulation, especially during dental examination and treatment [11,12,14,15]. Other barriers such as reduced energy levels, competing health priorities, distrust of dental providers, and reduced awareness of the presence of oral health problems have also been shown to impede access to dental services [16,17,18].

Additionally, systemic issues such as cost and accessibility can be further deterrents for many people in accessing dental services [19,20]. In a review of accessibility and affordability of healthcare across developed nations, 21–32% of adults in affluent countries, including the United States, Canada, France, and New Zealand, avoided dental care due to cost [20,21]. As noted in a report by the Australian Institute of Health and Welfare, a significant proportion of the Australian population experienced financial difficulty accessing dental services, with up to 57% of individuals covering a significant financial proportion of the “out of pocket” expenditure on dental services [22]. Those without access to private health insurance faced lengthy public system waitlists as an alternative [23].

By way of mitigating the strain on public wait lists, dental costs, and accessibility, research has explored the role of non-dental health professionals in providing oral health promotion to vulnerable populations [24]. Non-dental health professionals, including midwives, nurses, and allied health professionals, are effective in promoting oral health among priority populations such as those with chronic illness [25,26,27,28,29,30]. This is significant, as non-dental health professionals provide a means for early intervention screening or preventative education due to their close contact either by inpatient or community/outpatient service [26,27,28,29,30].

Similarly, dietitians have been shown to assist in early intervention and identification of oral health concerns [31]. In a recent scoping review undertaken by the study investigators, dietitians were noted to be knowledgeable in key areas of oral health including nutrition-related factors such as the impact of high cariogenic diets and risk-associated behaviours [31]. Further, dietitians were also routinely incorporating oral health promotion in their practice [31]. However, to our knowledge, no studies exist globally that explore the role and perceptions of dietitians in providing oral health promotion to clients in an ED clinical setting [31]. Hence, this paper aimed to explore the perceptions and role of dietitians in providing oral health promotion to their clients in an ED clinical setting. The specific objectives included:Understanding dietetic practice around oral health promotion in the ED clinical setting;The impact of ED-related oral health concerns based on clinical experience level;Understanding the barriers and facilitators to oral health promotion in practice.

## 2. Materials and Methods

### 2.1. Design

Given the dearth of information in this area, a descriptive qualitative approach was adopted to gain a more robust understanding of the relationship between dietitians and oral health promotion in the ED setting. This study forms part of a larger sequential multi-phase mixed methods study, which is guided by the “development phase of the Medical Research Council (MRC) framework” [32,33]. The MRC framework is a complex framework that is often used in health service delivery. Other phases of the larger study include a scoping review [31], a national survey of dietetic practice regarding oral health promotion in Australia, and the perceptions of individuals with an ED regarding oral health. The findings of these additional phases will be published elsewhere.

### 2.2. Recruitment

A combination of purposive and snowball sampling was used to recruit participants. Dietitians were recruited via two separate methods using an opt-in approach.

In the first method, dietitians who participated in the earlier phase involving a national survey and who worked in an ED clinical area were invited to take part in an interview by leaving their contact details for the research team.

In the second recruitment method, a publicly available database of ED-specific allied health professionals, available through a key professional organisation in this area (the Australia and New Zealand Academy for Eating Disorders), was reviewed for potential participants. Relevant dietitians were contacted via email with an opt-in approach to participate in a semi-structured online interview following a brief introduction to the study via information sheets. Interested participants were asked to send back their consent form with their availability for a semi-structured interview via video/teleconference.

### 2.3. Inclusion and Exclusion Criteria

To be eligible for inclusion in the semi-structured interviews, participants had to meet the following criteria:

Be recognised as a dietitian in Australia through a certified tertiary education course accredited by Dietitians Australia, including 20 weeks of clinical placement, and meet the requirements of the National Competency Standards to hold the title or meet the eligibility to be classified as a practising dietitian [34];
Currently working as a clinical dietitian in an ED-specific clinical setting or with clients who have an ED in any public or private inpatient or outpatient setting in Australia;Over the age of 18 years;Able to provide informed consent for participating in qualitative semi-structured interviews.

Individuals who did not meet the above criteria were not eligible to participate. Ethics approval for this study was sought from the Western Sydney University Human Research Ethics Committee, approval number: H13316. For anonymity, all participants were assigned a pseudonym before analysis. To assist with anonymity, individual interviews, rather than a focus group, were chosen to maintain privacy and confidentiality.

### 2.4. Data Collection

An interview guide was used to inform the semi-structured interviews. This allowed for the lead investigator to direct the conversation yet provided flexibility to follow the lead of the participant as needed. The interview guide was informed by the earlier scoping review [31] and the preliminary findings from the national survey of dietitians. The interview questions focused on the perceptions and practices of dietitians regarding oral health promotion as well as barriers and facilitators to oral health promotion in ED clinical settings (see Appendix A).

Additional demographic information including age, gender, highest education qualification, years of working as a dietitian, years of work in an ED clinical setting, and geographical location were recorded. Interviews were conducted by the lead investigator and associate investigators who are experienced in conducting qualitative interviews. All interviews lasted approximately 45 min to 1 h and were completed via video conferencing (Zoom) at a convenient time for the participants. Audio from the interviews were recorded with consent from the participants. The audio recordings were automatically transcribed into text and then professionally cleaned.

Recruitment continued until theoretical data saturation was reached. Data saturation was deemed met by the researchers as no new information surfaced and further coding and categorisation was no longer feasible [35,36].

### 2.5. Data Analysis

Once the transcription files were cleaned, they were imported into qualitative analysis software N*vivo for analysis [37]. A hybrid approach to analysis was undertaken using both deductive and inductive reasoning. The deductive approach was guided by the semi-structured interview guide and analysed through the lens of the Medical Research Council (MRC) framework. Data specifically identifying barriers and facilitators to the proposed intervention for both dietitians and clients formed the basis for the development of a priori codes. An inductive thematic approach allowed for the researcher to be guided by the participants’ voices and this is illustrated by the generation of salient themes and subthemes. This method of analysis is noted to be particularly useful in under-explored areas such as this topic [38,39]. As oral health promotion for dietitians working in ED clinical settings is thought to be novel, a thematic analysis of the data described the participant’s reality using their own written or spoken account [35]. Words, sentences, or paragraphs that captured the essence of the participants’ meaning were categorized. The categories and the frequency of recurring codes within them were used to develop the themes from the data [35].

### 2.6. Rigor

Many strategies were employed to enhance the trustworthiness of the overall concurrent data collection and analysis process. To ensure credibility, the lead investigator guided the recruitment process and assisted with organising interview times; all interviews with participants were led by the lead investigator, as lines of rapport, communication, and trust had already been built. Additionally, weekly meetings were held between the lead investigator and co-investigator (AG) to debrief, review, and prepare for subsequent interviews until theoretical saturation was reached. For dependability and to ensure the accuracy of verbatim transcriptions, the transcripts were reviewed by a professional transcription agency and re-reviewed by the lead investigator. Member checking of transcripts was not deemed feasible for participants due to the time constraints of their clinical workload.

Furthermore, the lead investigator reviewed all interview transcripts, and each co-investigator reviewed a sample of transcripts. Each conducted a thematic analysis independently, and then a coding meeting was held to discuss and confirm the coding structure and preliminary findings.

For transferability, detailed information on the recruitment and interview process, analysis, and findings including direct quotes from participants have been provided. Finally, the Standards for Reporting Qualitative Research (SRQR) [40] was used to guide the reporting of this qualitative study.

## 3. Results

### 3.1. Characteristics of Participants

A total of 14 dietitians (female: *n* = 13; male: *n* = 1) from several different states and territories in Australia, including Queensland (*n* = 2), New South Wales (*n* = 7), the Australian Capital Territory (*n* = 1), Victoria (*n* = 2), South Australia (*n* = 1), and Western Australia (*n* = 1) participated in this study. Dietitians were aged 26–60 years old and had, on average, 13 years of experience as clinical dietitians and over 9 years of total experience working in an ED clinical setting. Half (*n* = seven) were employed part-time in their clinical role. One dietitian had a doctoral qualification, and the majority had a Master’s degree (*n* = six, 43%). For anonymity, dietitians were assigned pseudonyms and will be referred to as Dn (where D = dietitian and *n* = participant number).

Following the inductive and deductive hybrid analysis, major themes were identified (Table 1). Participants discussed their knowledge of oral health specific to eating disorders, their current screening, education/counselling, referral practices, barriers, and facilitators to incorporating oral health screening and promotion in practice. Please refer to Appendix B for a definition of terms relating to the roles and practices of dietitians.

### 3.2. Knowledgeability of Eating-Disorder-Related Oral Health

#### 3.2.1. Impact of Poor Oral Health: “…their teeth are discoloured, they’ve got some erosion…”

Generally, participants had varying levels of understanding of general oral health and the specific oral health implications of eating disorders. Although a few reported that “*I don’t really know to be honest…risk factors for poor oral health, yeah I don’t know*” (D6) or “*I couldn’t tell you…*” (D10), most dietitians had some knowledge about the impact of poor oral health on overall health. This included the impact on their client’s chewing ability, variety of foods they were able to consume, and the quantity of food they were consuming: “[poor oral health] *decrease[s] their nutritional intake purely cause they don’t eat*” (D4):

“*…they’ve* [clients with an eating disorder] *got pain that is affecting the amount they are eating, as well as the diversity of foods they’re eating…*”(D2)

“*Oh it’s hugely* [regarding importance of oral health]. *First of all, if they’ve* [client with an ED] *got poor oral health then eating foods like fruit and vegetables and things that actually require crunching and chewing is incredibly difficult…*”(D5)

However, only some participants (*n* = five) had a deeper understanding of the impact of oral health on Eds. Those that were more knowledgeable had either worked in the dental profession or had close family working in the dental health services industry. These participants were able to identify “*the damage to the enamel from the acid, red gums…due to constantly bathing the teeth*” [referring to the teeth being bathed in acid during vomiting] (D7), as well as other signs and symptoms:

“*I’ve had quite young clients that have said ‘I’ve had to get all my teeth replaced really because the purging has been that bad’*”(D3)

“*Definitely you could see that if someone’s got a long term* [eating] *disorder, that you can definitely see that their teeth are discoloured, that they’ve got some erosion, discolouration, Also, poor oral hygiene…*”(D7)

They were also aware of the long-term implications of poor oral health on their clients in terms of the aesthetic appearance of teeth and diminished self-esteem and quality of life:

“*If that’s painful, I guess it’s going to bring down their quality of life and make it feel uncomfortable to keep up with general dental hygiene*”(D12)

“*She reported her dental symptoms like her sensitivity and how she just has no confidence with her smile*”(D3)

One participant highlighted her understanding of poor oral health, its link to co-morbid chronic disease, and the impact of malnutrition due to an ED:

“*…They’ve* [clients with an ED] *lost teeth, or having root canals, but they are probably not realising that they’ve got gum disease…they’ve got higher risk anyway with medications, especially with diabetes and cardiovascular disease or eating disorders, they’re all linked to poor oral health as well… they’re* [dietitians] *going to be concerned about oral health in eating disorders just because of malnutrition…”*(D5)

#### 3.2.2. Awareness of Current Guidelines and Referral Pathways: “*I don’t know if they exist or don’t exist*”

Overwhelmingly, most participants were unaware of any specific guidelines or policy for informing dietetic practice around oral health screening and promotion: “*No… I’m not aware of any* [guidelines] *specifically for dietitians*” (D3). Dietitian 1 exclaimed: “*I don’t know if they exist or don’t exist*”. Few dietitians were able to recall other oral health guidelines for practice that were non-specific to dietitians, such as “*through the ADA* [Australian Dental Association]…” (D7) and other local organisations or private practices:

“*Not any of the bigger picture guidelines, we have some local stuff that’s part of one of the clinics I work for…*”(D2)

Only one participant identified “*the oral health and dietitian position statement*” (D5), a national policy statement made following the collaboration between major dental health services and the peak national dietetic body; however, they felt it was inadequate. Similarly, nearly half of all participants were unaware of existing referral pathways to dental services for their clients.

“*No, I don’t know of a* [referral] *pathway. If it is, it would be word of mouth…I know of my own dentist. I might just say why don’t you check in with them?*”(D10)

### 3.3. Clinical Practices Relating to Oral Health

#### 3.3.1. Oral Health Promotion: “*Try to work on stopping the symptoms before it’s [too] late*”

Most dietitians reported providing oral health promotion or counselling at some point to their clients. Oral health promotion (OHP) came mostly from a nutrition-focused standpoint, such as managing oral health to reduce the risk of malnutrition, and focused on areas such as “*…brushing at least twice per day, watching your acidic foods… food sticking to your [teeth] especially dried fruit or anything that sticks*” (D4).

However, there was no consistency regarding when OHP was provided:

“*But* [it’s] *not something I enforce regularly, no… it’s not something that I am getting into every time I see them*”(D10)

“*I probably won’t go down that path very often…*”(D12)

Those with a previous connection to the dental field were very proactive in this area and oral health was often in the back of their mind: “*I always make sure I address it*” (D4).

A few dietitians working in subacute inpatient settings also took the initiative to ensure that texture modification of their client’s diet was part of their practice when considering oral health concerns:

“*We were so involved in the food service team we would look at how it could be modified. Choosing soft foods and those sorts of things. If someone has braces…if those issues were related to their eating disorder or separate from that we just helped them manage by varying the food.*”(D3)

Another dietitian commented that they “*…made sure that it* [food] *was soft enough and that the nursing staff knew* [about any oral health concerns]…” (D12) to mitigate issues such as ill-fitting dentures following client weight loss, pain, or sensitivity.

Some dietitians reported they would provide OHP only if it was raised by their client or if they thought they were at risk of dental problems like dental caries.

“*I would say, rarely. I think occasionally with patients who are—I would say it’s mainly in relation to the purging…Try to work on stopping the symptoms before it’s [too] late. I think that’s kind of more of the focus…*”(D1)

Providing written OHP resources was the most popular method cited for educating clients. Dietitians used resources that were developed by government organisations, professional bodies such as Dietitians Australia, or by their own practices. Topics often covered by the resources included those on care after vomiting, reducing acid wear, and sugar: “*I do one on sugar [oral health promotion] so they can see how much sugar is in things…I really reinforce the importance of oral health.*” (D4)

#### 3.3.2. Undertaking Screening: “*…it’s part of my clinical judgement*”

More than two-thirds of dietitians reported they were screening for oral health concerns as part of their clinical practice. Of these participants, few were undertaking it regularly with their clients—“*Yeah we pick it up* [oral health concerns] *when we assess them. We have it on our assessment form.*” (D5). More than half undertook screening on an ‘ad hoc’ basis or if their client raised any oral health issues.

“*Probably about seventy-ish percent of young people, I would ask a question about oral health…*” (D2)

“*I might ask fifteen to twenty percent of people about how their dentition is out of the one hundred percent that I see…But I don’t ask the question, to begin with.*”(D10)

Dietitians who were screening did not follow any set questions and were often guided by their clinical judgement.

“*…it’s part of my clinical judgement. I know what’s going on medically for this person that’s part of my skill set as a dietitian to know when I need to ask those questions*”(D2)

Screening questions that were asked included oral hygiene practices, presence of any pain, sensitivity, dental check-ups, and dietary choices:

“*I would ask them something along—‘do you brush your teeth? Do you find you’re sensitive to anything in your mouth?…*’”(D10)

“*Things like just asking them their hygiene…are they brushing their teeth? Are they noticing any pain? Are they going to the dentist regularly?*”(D3)

#### 3.3.3. Providing Referrals: “*But I’m doing that very seldomly. I’ve probably told one person*”

Generally, most dietitians were not providing referrals to their clients mostly due to limited knowledge of appropriate referral pathways, lack of appropriate trauma-informed dental professionals, and OHP being considered a lower priority due to other competing clinical demands.

“*No. Like I said, because my friend works locally in the area I practice in, I might say, if you’re looking for one—if they ask me, ‘do you know of one’? I might go suggest one. But I’m doing that very seldomly. I’ve probably told one person.*”(D10)

Additionally, a few participants noted that they were weary of making referrals to dentists who were not familiar with working with individuals with an ED or not trauma-informed as they felt this may negatively impact the care of their client:

“*I’m concerned that I’m going to send them to a dentist that is going to have no understanding of eating disorders and its very shaming…I get quite particular about who I’m going to send my clients to because otherwise, it’s going to erode all that beautiful work that I’ve done with my clients…I don’t need to send my patient that’s in a higher body weight who’s purging and for someone to make the comment and say ‘Well you don’t look like you’ve got an eating disorder*’”(D7).

### 3.4. Attitudes towards Promoting Oral Health

#### Acceptability: “*dietitians are a well-placed profession*”

Many participants recognised oral health as a contributor to overall health and as part of their scope of practice. Dietitians’ acceptability of oral health as part of their scope of practice and as an area that they should be familiar with was important as it helped create links with the impact poor oral health can have on their client’s overall health and wellbeing.

“*At the end of the day, oral health is actually important. It can also be a very expensive problem for people. It’s something that you would want to manage before it becomes a big problem… I think it would be important to understand what are the risks and why, and how to educate people around that in basic terms and be able to refer them on*”(D6)

Further, nearly all participants felt that “*dietitians are a well-placed profession to be doing that sort of screening*” (D2):

“*I think we at least have a role and helping people like even consider that* [oral health may be] *a problem*”(D14)

Many dietitians also went on to describe that they felt providing OHP, screening, and referring their clients was a part of their scope of practice: “*I think our role as dietitian is to connect that diet-disease relationship in any situation right, so if there’s going to be…an oral health issue that impacts eating either way we need to know about it…*” (D14)

A few participants who had prior experience or connections to the dental setting felt confident in performing OHP and screening: “*I’m confident I can do it…I have the knowledge to provide support…*” (D9). Specifically, these participants felt capable of providing fundamental oral hygiene advice “*like brushing your teeth, making sure that you’ve not had too many sweetened beverages…Trying to drink a lot of water…*” (D5), and had positive perceptions about providing nutritional support and asking screening questions: “*I would feel okay about asking questions or screening about it…*” (D11) and “*I’m confident that I can adjust any food and have an understanding of what food would be required to sort of help…*” (D12).

A few participants felt like their clients would be accepting of them providing OHP “*…whenever it has come up this hasn’t been met with any resistance*” [referring to the dietitian providing OHP advice to their client] (D8). They believed clients trusted their advice, and felt they would not judge them:

“*…they would be more comfortable with it* [screening]*…this is just a set of questions that gets asked to every person who walks through this room…might take some of that judgement away*”(D2)

“*…you know the dentist is going to judge me* [dietitian providing client perspective], *so I think we* [dietitians] *have quite a role in the conversation around helping people to get to an oral health professional*”(D14)

### 3.5. Barriers

#### 3.5.1. Competing Priorities: “*I need to just keep moving*”

Time constraints were identified as a barrier by nearly half the participants. Dietitians reported finding it hard to provide OHP or screening to at-risk clients when dealing with strict appointment times and competing health priorities.

“*Making sure that your appointment runs to 30 min or 60 min or whatever you’ve got…my barrier is probably something like time and waitlists*”(D2)

Competing priorities were also a barrier raised by a few participants. One dietitian noted this was because clients did not raise any oral health concerns with them and felt that it was a lower priority in their practice: “*To be honest, I don’t really go into it after that. I don’t say, by the way, if you have really sweet things it’s going to cause you dental caries. I have so many others* [Other clinical priorities] *that I want to get through them that I need to just keep moving*” (D10)

“*…I guess it* [screening] *would be a thing that… just doesn’t really come up that much… I guess because it doesn’t come up so much, it doesn’t come to my mind…it might not seem like a priority when I’m trying to get a lot of other information*”(D6)

#### 3.5.2. Client Barriers: *“shame is like a huge part of eating disorders*”

Generally, dietitians were very aware of challenges their clients may face such as shame that may deter them from seeking dental services or following through with a referral. As noted by one dietitian: “*shame is like a huge part of eating disorders, so if they think there’s any risk of being shamed and then I’d say that probably decreases the interest in like going along to actually see a dentist*” (D13) and “*I guess clients may find it uncomfortable or difficult to talk to a dentist about their eating disorder to see a new person and talk to them about their symptoms…*” (D6)

Further, some participants also raised concerns regarding challenges such as the cost that their clients may face when accessing dental services and highlighted this as a barrier to them instigating the conversation of OHP and making dental referrals: “*I think we have to be mindful of socioeconomic status and making sure that we’re not saying ‘oh you didn’t go to a dentist’ and someone can barely get food on the table*” (D12).

#### 3.5.3. Inadequate Training: “*I really need to refresh it before I feel confident doing it*”

The majority of participants felt that they had a basic awareness of oral health and hygiene. Participants expressed a “… *lack of awareness. Lack of confidence…*” (D9) and therefore, if required to provide more specific OHP, they would need further training.

“*I really need to refresh it before I feel confident doing it* [providing oral health counselling and screening].”(D8)

“*I guess the lack of training and education that I’ve had, and probably as well as talked about complete lack of knowledge around where people can go…bar looking up like Googling a dentist near you…I think without knowing much about the issue or what we can do, I don’t feel particularly confident talking about it*”(D6)

Lack of access to specific training around OHP for dietitians was another barrier. Nearly all dietitians reported mixed levels of access either having received training “*…from uni lectures years ago…*” (D10) or “*… it wasn’t really touched upon at uni…*” (D3) to having received OHP experience through sharing a workspace with a dental health professional. Dietitians were able to recall receiving information on areas such as general hygiene; however, the general sentiment was that training was inadequate and rare.

Only a couple of dietitians had received some general informal training via “*a widespread in-service for all community mental health staff…so in-house, but also very widespread and not specific to dietitians*” (D2).

#### 3.5.4. Resources: “*I think I’ve used some…in the past*”

Similarly, dietitians also had limited access to resources. Some dietitians were not aware of any resources about oral health complications associated with an eating disorder. For the participants who were using resources, they most often chose resources *“developed in our service…I think I’ve used some DA* [Dietitians Australia] *ones* [resources] *in the past*” (D1) or from other trusted organisations. A few participants had made direct relationships either personally or professionally with dental health professionals to be able to obtain resources and information. As described by one participant *“We’ve done a lot of health promotion work with a dentist at* [X] *dental hospital. It’s really her that made me aware of it…*” (D4).

#### 3.5.5. Lack of Guidelines for Practice: “*It was a good starting point…*”

The position statement between the national dietitian’s organisation in Australia and Dental Health Services Victoria (a state in Australia) was identified by one participant; however, this participant felt the statement was lacking more practical suggestions for dietitians and missed information on how dietitians and dental services could work collaboratively in this space:

“*It* [Position statement] *wasn’t very practical. It was a good starting point…but it needed more work…What our* [dietitian] *roles were and how you work together* [dietitian and dentist], *and maybe some suggestions of how you put into practice in terms of what questions you are going to ask, a flow chart of what could happen with working with each other…*”(D5)

### 3.6. Facilitators

#### 3.6.1. Collaborative Practice: “*that was great to make that contact with her…*”

A couple of participants also highlighted that using their initiative to engage in collaborative practice with dental services assisted in facilitating the identification of potential referral pathways and promoted open dialogue for the flow of knowledge of oral health from the subject matter experts (dental professionals).

“*I went to the dentist myself a week ago…she had a particular interest in dental erosion…she’s quite close to my practice as well so that was great to make that contact with her, that she’d be able to care for my patients…*”(D7)

“*I think the director* [of the nutrition private practice] *then went ‘okay, we need a dentist’ so she went found a dentist, educated them and so she was quite motivated to build connections in the community, with people who could understand eating disorders a bit better, so now we can safely refer patients there*”(D14)

#### 3.6.2. Model of Care: “*there needs to be dietitians involved…*”

Unanimously, participants reported that training was required to support their knowledge and help them feel confident: “*I suppose when you understand why something is, it makes you feel more comfortable to be able to talk to people about it*” (D6). Specifically, most participants identified that online training that was either provided as a live webinar session or self-paced learning modules would be most beneficial to them. Less than half preferred face-to-face style training. Most participants felt that “…*a half-hour kind of session. You can do it in your own pace as well and at your own time. You can do it early in the morning, during the day…*” (D4) would be sufficient; however, a couple of participants did highlight that more complex training would be better face-to-face.

In terms of the knowledge, many wanted to gain a deeper insight into the theory around the importance of oral health in eating disorders, risk factors for clients, techniques, and the role of the dietitian in oral health promotion:

“*…What’s the scope of the dietitian? Make it very clear what I am going to do… I need some strategies because I might not be able to get my patient to the dentist so I’m gonna need to know some recommendations… ‘what are the products available?’… ‘what are my options?’*”(D7)

A smaller number of dietitians also felt that providing training at the undergraduate or master’s level would assist future dietitians in equipping them with skills for oral health promotion and encourage it to become part of standard dietetic practice.

“*I think it needs to be touched on and explained. To me, it was one lecture, but at least it was touched on…it’ll be really good to have especially in your masters where you’re starting to do pracs* [practical training]*…where you refer to and what you can say, like quite simply, ‘do you brush your teeth?’, ‘Are you flossing?’, ‘When did you last see your dentist?’…*”(D4)

All participants identified a need for additional resources in their practice. Some either recognised they required resources for themselves to guide screening or provide referrals. Specifically, participants wanted a resource that was “*real black and white…*” (D2) detailing education for maintaining oral hygiene and clear referral pathways to dentists. These participants also preferred this to be provided as “…*just a written resource, like a little training video…*” (D1), “*just a page handout…what are the most important questions to ask clients? The rationale as to why…*” (D3). They also felt as though resources should be developed in consultation with dietitians and *“maybe approved or like in collaboration with like the dental association…like a joint initiative…*” (D13) to ensure resources are appropriate:

“*It would be good if dietitians were involved in producing some of that* [resources]*…some of the information they produced was quite concerning with the type of foods…how much sugar’s in food like they’re putting on a poster how much sugar is in plain milk…there needs to be dietitians involved in producing their education material, especially around nutrition…*” (D5)

Other dietitians identified resources that they would require for their clients. For the most part, participants wanted resources that provided information on risk factors for oral health around purging and nutrition: “*it’s on one piece of A4 paper, typically, it’s just one sheet and it’s a summary of the important things and what you need to do…*” (D9).

## 4. Discussion

This study aimed to investigate the perceptions and role of dietitians in promoting oral health in the ED clinical setting, as well as the barriers and facilitators. Being the first study of its kind world-wide, the findings from this phase of research built upon an earlier scoping review [31] by addressing current gaps in the literature around Australian dietitians toward promoting oral health. Most studies to date have focused on the role of the dietitian in promoting oral health in other vulnerable populations [41,42,43,44]. There have been no investigations into the role of dietitians in promoting oral health in eating disorder clinical settings.

A basic level of understanding and overall lack of confidence in promoting oral health was reported by most dietitians and, consequentially, this was reflected in their seldom practice of oral health promotion (OHP). Consistent with other research looking at dietitians promoting oral health in population groups, such as women, infants, children, people with HIV, and the elderly, dietitians were all shown to have limited knowledge and confidence [31,41,42,43,44,45]. Similarly, in another study exploring OHP by allied health professionals in an Australian stroke rehabilitation setting, allied health team members, including physiotherapists and speech pathologists, were identified as key members that had the capacity and opportunity due to their close clinical relationships with patients to be providing OHP [25]. Similar to the dietitians in this study, allied health members reported lack of knowledge, training, and resources as barriers to promoting oral health practice [25]. From these studies, when non-dental health professionals feel like they had limited knowledge or confidence, they were less likely to incorporate OHP as part of their standard practice [25,31]. The confidence of non-dental health professionals and specifically dietitians cannot be improved if there is no training or resources to support promoting oral health in their practice.

Understanding the missing piece to improving dietitian confidence and capacity was illustrated by the dietitians who had prior training, work experience, or professional connections within the dental arena. It was these same dietitians who felt confident enough to seek out collaborations with dental care providers to bridge the gap between professions and allow for knowledge transfer. It must be noted that this is not a new finding. Dietitians in other vulnerable population groups who received training and resources moved on to competently incorporate OHP into practice. In a study by Biordi et al., 2015, a nurse practitioner–dietitian program was established targeting children from low-income areas for basic oral assessment and education [46]. Following adequate training, dietitians were able to competently work in the nurse–dietitian dyad towards promoting oral health, specifically, nutrition-focused oral health education for their clients and referrals to dental professionals. These studies suggest that when dietitians are supported with training or access to resources, they more readily engage in promoting oral health.

Another key finding from this study was the acceptability of oral health as part of dietetic practice, despite most dietitians being unaware of the existing national and international guidelines for practice [47,48]. This finding was in alignment to other studies with dietitians working across different clinical settings. In a review study led by Brody et al., which involved providing OHP training to dietitians working in an aged care setting, participants showed an acceptance of OHP as part of their role and intent to include OHP in their practice despite the challenges of limited knowledge and training [45]. A similar recognition of the role of dietitians in OHP was found in student dietitians following a dentistry clinical rotation [49]. These findings are significant as it shows there is a space for oral health in dietetic practice. Acceptability of non-dental health professionals in providing pre-emptive OHP is essential due to the implications it has for early intervention in vulnerable client groups and the potential for reducing the necessity for costly intervention methods and extensive public system wait lists, as noted by other studies where non-dental health professionals have undertaken OHP [50].

Considering the acceptability and capacity of dietitians to promote oral health, significant recurring barriers were identified. None more so than the lack of training and resources for clinicians. As with any non-dental health professionals trying to expand their capacity in oral health, this was an expected finding and, again, one that had been foretold in previous studies. As reported by Lieffers and colleagues in a study of nutrition care practices around oral health conditions of dietitians and dental professionals, inadequate training was listed amongst the three most common barriers reported by dietitians [51]. This is also supported by the findings from a review of dietetic practice where no formalised training for OHP was identified and there were limited dietetic-specific resources for upskilling in OHP [31].

Additionally, the lack of practical advice from guidelines or recommendations was another barrier. Both national and international recommendations lack clear steps or guidance towards implementing OHP into practice [47,48]. Furthermore, in another guideline specifically focusing on the practice standards for dietitians working in ED clinical areas, oral health risk and mitigation strategies were not covered [52]. The lack of clear direction or advice provided in these guidelines is similar to the dearth of information on oral health in undergraduate dietetic studies. As highlighted in the findings, most participants had little to no recollection of studying oral health in their tertiary education. Guidelines and recommendations in practice are essential to ensure patients are receiving quality evidence-based holistic care [53]. Importantly, they are also essential in clinical decision making [53]. Hence, acknowledgment that promoting oral health is required in dietetic practice is not sufficient if measures for supporting the practice of oral health promotion such as training or practice recommendations are not available.

In saying this, the barriers identified in this study highlight key target areas that need to be explored further or developed prior to implementation into practice. Barriers often encountered, such as competing clinical priorities, are salient findings when engaging non-dental health professionals in OHP; however, the barriers can be negated with effective implementation through a co-design model where dietitians can be involved in developing an acceptable and feasible model of care for OHP [54]. The need for specific oral health training of non-dental health professionals in vulnerable populations is not unique to this study, but a finding echoed by most non-dental health professionals [28,30,55,56,57]. When non-dental health professionals are engaged in co-design of models of care, they are more seamlessly implemented and accepted [54]. To provide a framework for a model of care, recommendations developed by key dietetic bodies should be reviewed to ensure recommendations are pragmatic and can be used to guide the implementation of OHP into practice. Furthermore, a short, validated screening tool, similar to an oral health screening tool used by dietitians in the HIV clinical setting [58], would facilitate practice, and resources for dietitians that complement ED client oral-health-specific resources should be developed [59]. Additionally, it is essential that the barriers limiting individuals from seeking out preventative dental services should be considered [60,61]. Participants in this study hypothesised that the lack of available and accessible trauma-informed dental providers and cost may deter clients from visiting their dentists. In other populations at risk of poor oral health, the biggest predictors of engagement with dental services were awareness of their vulnerability to poor oral health and cost [18,62]. Further, service constraints in Australia and other developed nations including lengthy wait lists or financially inaccessible private dental care will continue to be barriers, especially to the vulnerable population groups [20,22,23]. However, as demonstrated in an economic evaluation of midwives promoting oral health in pregnancy, using existing referral pathways and increasing awareness amongst clients was found to be cost effective in the short term despite client barriers [50]. Thereby, working with dietitians to develop and implement a brief screening tool and utilising existing referral pathways could provide a method for reducing cost by raising awareness and championing early intervention referrals to dental services. Lastly, further research is required to understand the experiences and acceptability of clients with an ED towards receiving OHP from dietitians and current barriers towards engaging in dental health services.

### Limitations

Although this is a developing topic providing novel insights into the requirements needed for capacity building dietitians to promote oral health, there are some limitations that should be acknowledged. The majority of the dietitians who were interviewed practiced in a metropolitan setting. With minimal representation from dietitians providing services to rural or remote population groups, it is difficult to extrapolate these results to understand the differences in practices, and if additional supports are required. Considering the COVID-19 pandemic, all interviews were conducted via video conferencing and, hence, that may have impacted the depth of questioning or lines of inquiry that may have otherwise been captured if interviews were completed face-to-face. Additionally, three dietitians who participated either had qualified and practiced as dental assistants or had a spouse who was part of the dental profession. Although these participants influenced the findings, they were also well-placed as an example for how knowledge of oral health can assist dietitians in promoting oral health and screening within their practice.

## 5. Conclusions

Dietitians have an awareness of how oral health can impact their clients experiencing an ED. However, despite dietitians acknowledging OHP is part of their role, barriers including lack of resources and training impede their ability to play an active role in this area as part of their standard practice.

To facilitate OHP in ED clinical practice, it is not adequate to simply acknowledge that dietitians should be performing this role. Clear guidelines with recommendations for practice that outline how OHP can be implemented need to be developed. Further, resources and training that support non-dental health professionals in OHP need to include dietitians so that vulnerable clients such as individuals with an ED have the opportunity for access to early intervention.

## Figures and Tables

**Table 1 ijerph-19-14193-t001:** Major themes and subthemes.

Major Themes	Subthemes
Knowledgeability of eating-disorder-related oral health	Impact of poor oral health Awareness of current guidelines and referral pathways
Clinical practices relating to oral health	Oral health promotion Undertaking screening Providing referrals
Attitudes towards promoting oral health	Acceptability Barriers Competing priorities Client barriers Inadequate training Resources Lack of guidelines for practice Facilitators Collaborative practice Model of Care

## Data Availability

Not applicable.

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
