# Peer review of "Dietitians’ Experiences of Providing Oral Health Promotion to Clients with an Eating Disorder: A Qualitative Study"

_ijerph, 2022, doi:10.3390/ijerph192114193_

Round 1

Reviewer 1 Report

The manuscript needs more scientific input. 

I found a similar review article from the same group "Eating disorders and oral health: a scoping review on the role of dietitians". The article they submitted in your journal is more of a survey than any scientific approach.

Author Response

Feedback:

The manuscript needs more scientific input. 

I found a similar review article from the same group "Eating disorders and oral health: a scoping review on the role of dietitians". The article they submitted in your journal is more of a survey than any scientific approach.

Comment:

Thank you for your feedback. The scoping review titled: "Eating disorders and oral health: a scoping review on the role of dietitians" by the study investigators has been referred to in the background section of the manuscript as it provides the background evidence for this current qualitative study. The scoping review mapped the scope of available research on the topic of dietitians and oral health promotion in general. It highlighted the lack of evidence of ED clinical dietitians and oral health promotion. Therefore, the current manuscript aims to contribute to this knowledge gap. Our study used a descriptive qualitative approach and interviews as the method of qualitative data collection. We explored in-depth the perspectives of dietitians working in ED clinical settings regarding their knowledge, attitudes and practices around oral health promotion. As such, this qualitative study generated knowledge through the process of deductive and inductive reasoning of the data.

To make this clearer, ‘undertaken by the study investigators’ has been included to line 77 of the manuscript. We have also attempted to make it clearer in line 99, that the national survey and scoping review belong to the ‘larger’ sequential multiphase mixed methods study.

Reviewer 2 Report

The manuscript is entitled "Dietitians’ experiences of providing oral health promotion to clients with an eating disorder: A Qualitative study."

The manuscript is very interesting and has a suitable title, an informative abstract, and is clearly written in all the sections. Of course, the number of studied samples could be increased. 

I appreciated the high quality of the presentation of the manuscript; therefore, the manuscript can be accepted in its present form.

Author Response

Feedback:

Reviewer 2:

The manuscript is entitled "Dietitians’ experiences of providing oral health promotion to clients with an eating disorder: A Qualitative study."

The manuscript is very interesting and has a suitable title, an informative abstract, and is clearly written in all the sections. Of course, the number of studied samples could be increased. 

I appreciated the high quality of the presentation of the manuscript; therefore, the manuscript can be accepted in its present form.

Comment:

Thank you for your feedback.

Reviewer 3 Report

Nicely performed and reported qualitative analysis on dietitians' perceptions about providing oral health promotion to their clients in an ED clinical setting. A few comments:

In the introduction, and starting from line 58, the authors bring up the accessibility issue. However, an interconnection phrase highlighting the ED population's vulnerable feature is missing to make a smooth transition from the previous paragraph.

Consider moving the "Definition of terms" to the appendix. That would make the manuscript's main body thinner.

The extensive interview  results sectioned analysis can be resumed and eventually grouped by weaknesses  and strengths regarding dietitians' deducted/inducted oral health promotion skills in CONCLUSIONS

Regarding the interviewed dietitians, was there some blinding effort during recruitment, so they were unaware of the question's nature to avoid preparedness and bias related to unauthentic answers? Could this be a limitation?

Author Response

Reviewer 3:

Nicely performed and reported qualitative analysis on dietitians' perceptions about providing oral health promotion to their clients in an ED clinical setting. A few comments:

In the introduction, and starting from line 58, the authors bring up the accessibility issue. However, an interconnection phrase highlighting the ED population's vulnerable feature is missing to make a smooth transition from the previous paragraph.

Consider moving the "Definition of terms" to the appendix. That would make the manuscript's main body thinner.

The extensive interview results sectioned analysis can be resumed and eventually grouped by weaknesses and strengths regarding dietitians' deducted/inducted oral health promotion skills in

Regarding the interviewed dietitians, was there some blinding effort during recruitment, so they were unaware of the question's nature to avoid preparedness and bias related to unauthentic answers? Could this be a limitation?

Comment:

Thank you for your feedback.

We have changed the phrase used in line 60 to show the connection between the previous paragraph and continuation of discussion re: accessibility issues

Definition of terms has been moved into an appendix as suggest. Appendix B is now located at the bottom of the document line 714, prior to references. We have also directed readers to the appendix for the definition of terms in the results section. Please see line 208-209.

This is a valid suggestion, nonetheless, by following through with this would change the focus of this study. In saying this, under the major theme ‘Attitudes towards promoting oral health’, there are sub themes ‘barriers’ and ‘facilitators’ which covers the strengths and weaknesses to oral health promotion as identified by the study participants during interviews.

Thank you for your question. Participants had an awareness of the aims of the study based on the information provided in the participant information sheet. Prior to the interviews participants were not given a copy of the interview guide and therefore, were not able to prepare answers prior to the interviews taking place.  Therefore the participants responses were authentic and there was no bias.

Reviewer 4 Report

How many dietitians working in ED settings are in total in Australia? All 14 were trained in the same place? 

Conclusions should include only statements derived from the results. Consider re-phrasing the recommendations.

Check references!! 

Consider shortening the article to make it more 'readable'.

Author Response

Reviewer 4:

How many dietitians working in ED settings are in total in Australia? All 14 were trained in the same place? 

Conclusions should include only statements derived from the results. Consider re-phrasing the recommendations.

Check references!!

Consider shortening the article to make it more 'readable'.

Comment:

Thank you for your feedback. Given that it is not mandatory for dietitians to be registered by a governing body such as Dietitians Australia, it is difficult to determine how many dietitians are specifically working in ED clinical settings. All 14 dietitians were not trained in the same place. As noted in the demographics section, the 14 dietitians were from various states and territories across the vast geographical landscape of Australia and hence, would have completed their training in different locations.

The conclusion has been revised to include a summary sentence of the results . Please see lines 608-610. Thank you. References have been checked and the formatting error with journal names has been resolved. Thank you for your suggestion. Efforts has been made to make the paper shorter. This has included deleting an irrelevant section before Table 1 and reformatting Appendix A (reduced to half  its original length)

Round 2

Reviewer 1 Report

The manuscript has been significantly improved.